# Innovative Biomarkers for Obesity and Type 1 Diabetes Based on *Bifidobacterium* and Metabolomic Profiling

**DOI:** 10.3390/microorganisms12050931

**Published:** 2024-05-03

**Authors:** Angelica Nobili, Marco Pane, Mariya Skvortsova, Meryam Ben Salem, Stephan Morgenthaler, Emily Jamieson, Marina Di Stefano, Eirini Bathrellou, Eirini Mamalaki, Victoria Ramos-Garcia, Julia Kuligowski, Miltiadis Vasileiadis, Panagiotis Georgiadis, Marika Falcone, Paulo Refinetti

**Affiliations:** 1Autoimmune Pathogenesis Unit, Division of Immunology, Transplantation and Infectious Diseases, IRCCS San Raffaele Scientific Institute, 20132 Milan, Italy; angelicanobili@hotmail.com (A.N.); distefano.marina@hsr.it (M.D.S.); falcone.marika@hsr.it (M.F.); 2Probiotical Research, 28100 Novara, Italy; m.pane@probiotical.com; 3REM Analytics SA, 1870 Monthey, Switzerland; skvortsm@gmail.com (M.S.); meryam.bensalem@remanalytics.ch (M.B.S.); stephan.morgenthaler@remanalytics.ch (S.M.); emily.jamieson@remanalytics.ch (E.J.); 4Department of Mathematics, Ecole Polytechnique Fédérale de Lausanne, 1015 Lausanne, Switzerland; 5Department of Nutrition and Dietetics, Harokopio University, 17676 Athens, Greece; ebathrellou@hua.gr (E.B.); emamal@hua.gr (E.M.); 6Neonatal Research Group, Health Research Institute La Fe, 46026 Valencia, Spain; victoria.ramosgarcia@gmail.com (V.R.-G.); julia.kuligowski@uv.es (J.K.); 7Alpes Lasers SA, 2072 St. Blaise, Switzerland; miltiadis.vasileiadis@alpeslasers.ch (M.V.); panagiotis.georgiadis@alpeslasers.ch (P.G.)

**Keywords:** gut microbiome, *Bifidobacterium*, short-chain fatty acids, branched amino acids, obesity, type 1 diabetes, children, capillary electrophoresis

## Abstract

The role of *Bifidobacterium* species and microbial metabolites such as short-chain fatty acids (SCFAs) and human milk oligosaccharides in controlling intestinal inflammation and the pathogenesis of obesity and type 1 diabetes (T1D) has been largely studied in recent years. This paper discusses the discovery of signature biomarkers for obesity and T1D based on data from a novel test for profiling several *Bifidobacterium* species, combined with metabolomic analysis. Through the NUTRISHIELD clinical study, a total of 98 children were recruited: 40 healthy controls, 40 type 1 diabetics, and 18 obese children. *Bifidobacterium* profiles were assessed in stool samples through an innovative test allowing high taxonomic resolution and precise quantification, while SCFAs and branched amino acids were measured in urine samples through gas chromatography–mass spectrometry (GC-MS). KIDMED questionnaires were used to evaluate the children’s dietary habits and correlate them with the *Bifidobacterium* and metabolomic profiles. We found that *B. longum* subs. *infantis* and *B. breve* were higher in individuals with obesity, while *B. bifidum* and *B. longum* subs. *longum* were lower compared to healthy individuals. In individuals with T1D, alterations were found at the metabolic level, with an overall increase in the level of the most measured metabolites. The high taxonomic resolution of the *Bifidobacterium* test used meant strong correlations between the concentrations of valine and isoleucine, and the relative abundance of some *Bifidobacterium* species such as *B. longum* subs. *infantis*, **B. breve**, and *B. bifidum* could be observed.

## 1. Introduction

The gut microbiome, particularly human-related species of the *Bifidobacterium* genus, is emerging as a significant player in the pathogenesis of obesity and type 1 diabetes (T1D). T1D affects 0.13% of children in Europe [1], significantly limiting the quality of life of affected children [2].

The importance of the commensal gut microbiome composition and its metabolites in obesity and T1D has been clearly demonstrated. On the other hand, the metabolism modification associated with those conditions (e.g., high blood glycaemic value) can modify the microbiome composition and microbiome-derived metabolites [3].

In both obesity and T1D, the *Bifidobacterium* genus plays a clear role [4,5], with alterations in the relative abundance of different *Bifidobacterium* species evident in individuals affected by these conditions. *Bifidobacterium* is a well-studied genus that is commonly used as a probiotic [6] due to its clear anti-inflammatory effects and other beneficial properties.

From a human physiological viewpoint, specific *Bifidobacterium* species, i.e., human-related bifidobacteria (HRB), have been documented to play major roles in health, well-being, and metabolism, as they are involved in the production of essential vitamins, SCFAs, and other immune regulatory factors. The importance of HRB for numerous physiological conditions has been clearly demonstrated. For example, HRB are able to metabolise human milk oligosaccharides (HMOs), helping the anatomical, physiological and immunological development of the infant [7]. An association of HRB with mental disorders has also been suggested and documented [8].

Advanced Testing for Genetic Composition (ATGC) [9] is a technology platform for quantitatively profiling mixed genetic populations, even those that are taxonomically close. This platform was used to develop a new quantitative test for profiling several HRB. The test was challenged in a proof-of-concept study through the analysis of blind samples prepared by mixing pure cultures of well-characterised *Bifidobacterium* species. We associated this innovative test with metabolic profiling, diet evaluation (KIDMED), and disease condition (obesity/T1D) to investigate the relationship between the relative abundance of HRB, the release of different microbial metabolites, diet composition, and obesity/T1D. The results presented here are part of the NUTRISHIELD project [10] focusing on personalised nutrition. The long-term objective is to identify specific microbiome and metabolite profiles in children impacted by obesity and T1D to deliver personalised nutrition to improve the clinical management of those conditions.

## 2. Materials and Methods

### 2.1. Study Design

Male and female children aged 7–17 were recruited at the Paediatric Department of the San Raffaele Hospital in Milan, Italy. These children formed three groups, those diagnosed with obesity, defined as a body mass index (BMI) > 30 (n = 18), T1D (n = 40), and healthy controls (HCs) (n = 40). The diagnosis of T1D was based on the criteria of the American Diabetes Association (American Diabetes Association Professional Practice, 2022 #1358). Obese and T1D children were recruited at the time of diagnosis (first visit) or during a periodic visit. HCs were healthy children admitted to the San Raffaele Hospital for orthopaedic or surgical interventions. At the time of enrolment, the paediatricians informed the parents of the recruited children about the rationale and purpose of this study, the lack of reported risks related to the collection of stool and urine samples, the effort required to take part in this study, and their right to withdraw their consent at any time. Parents of the recruited children gave written informed consent, complying with the study procedure, and were aware that they donated stool and urine samples for research purposes. Stool and urine (10 mL) samples were collected by the participants at home, no more than 24 h prior to delivery to the hospital, and stored at 4 °C until being processed in our laboratory. Stool samples were placed in a fixative solution (95% ethanol, with 1 mM EDTA and 0.5% SDS), aliquoted in 2 mL vials, labelled, and stored at −20 °C. Urine samples were aliquoted in 1 mL vials, labelled, and stored at −80 °C until shipment. Sociodemographic information, psychological factors, lifestyle factors, and dietary assessments (FFQ) were also collected at the time of enrolment and dietary indices such as the Health Eating Index (HEI) and the KIDMED score (adherence to the Mediterranean diet) were extrapolated. This study was approved by the Institutional Ethical Committee of the IRCCS San Raffaele Scientific Institute (Protocol: NUTRI-T1D-2019). Individuals’ identifiable private information was protected according to the EU General Data Protection Regulation (EU-GDPR) with the help of the Institutional Data Protection Officer. The Clinical Research Investigator assigned a code to each patient and identifiers that link to protected health information.

The fixative solution was previously validated for its ability to (1) inactivate *Bifidobacterium* and prevent its growth (i.e., change in relative abundance), (2) maintain the DNA and keep it extractable, and (3) inactivate other microorganisms present in stool that could compromise the results. Validation was performed as follows:Pure *Bifidobacterium* cultures of all targeted species and subspecies were suspended in the fixative solution and kept at RT for 24 h. The solution was then centrifuged, the fixative solution was discarded, and the cells were placed back in the cultures. No growth was observed.DNA extracted from both pure cultures in fixative solution, as well as actual samples and spiked samples (i.e., stool samples spiked with a mix of *Bifidobacterium*), was prepared and analysed.A significant number of stool samples collected in the fixative solution were re-cultured to check for growth. No growth was detected.

These validation experiments were carried out in collaboration with IHMA Europe (Monthey, Switzerland), a CAP-accredited laboratory for microbiology.

### 2.2. Bifidobacterium Profiling in Stool Samples

The BifidoZoom service (REM Analytics, Monthey, Switzerland) was used to profile the *Bifidobacterium* species in stool samples. The test is based on ATGC [9], which relies on a workflow of Polymerase Chain Reaction (PCR) associated with cycling temperature capillary electrophoresis (CTCE) [11].

CTCE enables the identification and precise quantification of single nucleotide polymorphisms (SNPs) occurring in the fragments amplified by PCR. The assay was designed using REM’s bioinformatic primer platform [9]. In brief, it is possible to identify DNA fragment sets containing relevant SNPs, and thus quickly design complex assays. For example, to measure the relative abundance of *B. breve* and *B. longum*, fragment sets whose SNP differentiates between these species are required. Each primer is then optimised and validated using existing reference material (pure bacterial cultures) for sensitivity, specificity, and potential biases. The *Bifidobacterium* (sub)species that were measured are *Bifidobacterium longum* (subs. *longum* and subs. *infantis*), *B. animalis* (subs. *animalis* and subs. *lactis*), *B. bifidum*, *B. breve*, *B. adolescentis*, and *B. pseudocatenulatum*.

The specificity of the BifidoZoom assay components was tested using pure *Bifidobacterium* cultures (IHMA Europe, Monthey, Switzerland) fixed in ethanol. Individual fixed cultures were mixed in a combinatorial fashion to create all possible “pairs” (e.g., *B. bifidum + B. breve*, *B. bifidum*, and *B. longum*). The resulting mixes (and individual cultures) were tested with all primer pairs of the assay to check for specificity.

During the analysis of samples from the study, all samples were processed in batches of 20 samples. With each batch of samples, a negative control (i.e., a pure fixative without a stool sample), as well as a positive control, were used. The positive control was a stool sample that was spiked with a known mixture of all target organisms.

DNA extraction followed the BifidoZoom DNA extraction protocol (REM Analytics, Monthey, Switzerland). It includes sample homogenisation through bead beating, followed by the removal of humic acids using calcium chloride, and then DNA extraction based on precipitation with isopropanol.

### 2.3. Profiling Test of Mock-Up Mixed Bifidobacterium spp.

To demonstrate the precision and quantitative accuracy of the method, seven blind mixes, in which the relative abundance of each (sub)species was different, were analysed. Six pure cultures of freeze-dried strains, see Table 1, were prepared at Probiotical S.p.A (Novara, Italy). For each culture, the potency was determined using a fluorescence flow cytometry (FFC)-based method with the FACSCalibur instrument (Becton Dickinson, Franklin Lakes, NJ, USA) and the results were expressed as Total Fluorescent Units (TFUs). All lyophilised samples were preserved at −20 °C within airtight aluminium sachets until analysis. For reconstitution, a 1:10 (g/g) ratio was adopted, employing PBS. Samples underwent homogenisation using a stomacher (Seward stomacher model 400, 260 RPM, 4 min). Cell viability analysis utilised the BD Cell Viability Kit with liquid counting beads (BD Biosciences, Cat. no. 349483), with staining procedures adhering to the standards set forth in ISO 19344: IDF 232 (2015) [12] Starter Cultures, probiotics and fermented products. Quantification of lactic acid bacteria by flow cytometry.

In the staining process, 100 μL of the diluted microbial suspension, containing an estimated 10^5^–10^6^ cells/mL in buffered peptone water, was combined with 835 μL of PBS. Subsequently, 10 μL of propidium iodide (PI) at a prediluted concentration of 0.2 mmol/L and 5 μL of thiazole orange (TO) at 42 μmol/L were introduced. Following a brief vortex, the mixture was incubated at 37 °C for 15 min in a dark environment. Counting beads were vortexed for 30 s before adding 50 μL to the cell suspension to achieve a final volume of 1 mL. Microbial cells were identified using an SSC-H (Side Scatter) threshold, and a cell gate was established using forward versus side scatter (FSC-H vs. SSC-H) parameters. TO fluorescence was primarily detected in the FL1 channel, while PI fluorescence was predominantly observed in the FL3 channel. The optimal segregation of live and dead cell populations was achieved through an FL1 versus FL3 plot. To mitigate false positives and negatives, reference control gating was established using a fresh culture of *L. rhamnosus GG* to represent the live population, while a sample of the same culture treated with isopropanol served as a reference for the dead cell population. The live sample was singularly stained with TO, whereas the dead sample was stained with PI, ensuring accurate discrimination between viable and non-viable cells. The analysis was conducted by Biolab Research SRL (Novara, Italy) [13]. Seven mixes were then prepared, where the weight of each component was accurately measured. The first mock-up mix had an equivalent ratio in weight between all of the seven (sub)species, with a total weight of 120 g. Each of the subsequent six mock-up mixes had one single *Bifidobacterium* (sub)species increased by a factor of 5 (100 g) and all of the other (sub)species were reduced by a factor of 5 (4 g each) for a total weight of 120 g. Using TFU readings for each of the single inputs, the relative abundance of each strain was predicted and kept blind. The mixes were numbered and sent to REM Analytics (Monthey, Switzerland), who carried out the profiling using the BifidoZoom test. Measured relative abundances were reported before the composition was revealed to the analyst.

### 2.4. KIDMED Questionnaire

The KIDMED score was originally developed to combine Mediterranean diet characteristics as well as general dietary guidelines for children in a single index [14]. It was based on principles sustaining healthy, Mediterranean dietary patterns (e.g., daily fruit and vegetable consumption, and weekly intake of fish and legumes), as well as on those that undermine it (e.g., frequent fast-food intake, and an increased consumption of sweets). The index comprises 16 “yes or no” questions. Those denoting a negative connotation are assigned a value of −1 and those with a positive aspect +1. Total scores range from −4 to 12 and are divided into three levels: (1) ≥8, optimal Mediterranean diet; (2) 4–7, improvement needed to adjust intake to Mediterranean diet; (3) ≤3, very low diet quality. The index has been used in a variety of settings and countries [15].

### 2.5. Metabolomic Analysis in Urine

SCFAs (i.e., acetic acid, propionic acid, butyric acid, valeric acid, caproic acid, heptanoic acid, isobutyric acid, 2-methylbutyric acid, and isovaleric acid) and branched-chain amino acids (BCAAs) (i.e., valine, leucine, and isoleucine) have been measured in urine samples using GC-MS [16]. The results were normalised using creatinine concentrations and expressed as mmol/g creatinine. Creatinine was quantified following the manufacturer’s instructions of the DetectX^®^ urinary creatinine detection kit from Arbor Assays (Ann Arbor, MI, USA) based on the Jaffe’s method.

### 2.6. Statistical Analysis

Statistical analysis was performed using R (version 4.3.2). Unidimensional analysis and significance tests were performed using a bilateral Wilcoxon test, without correction for multiplicity testing. Multidimensional analysis was performed using the MixOmics R package (version 6.24.0) [17]. To perform partial least square discriminant analysis (PLS-DA), the standard PLS-DA function of MixOmics was used.

## 3. Results

### 3.1. Bifidobacterium Mock-Up Mix Profiling

In Figure 1, we observe a close alignment between the predicted values and the actual measurements, with an average discrepancy of approximately 6%. The average discrepancy was calculated as the average absolute difference between all predicted and observed values (the mean absolute error).

Biolab Research FFC analysis yielded a Reproducibility Standard Deviation (S_R_) for Total Fluorescent Units (TFUs) of 0.10 [log10 equivalent] complying with the ISO 19344 benchmark of an S_R_ of 0.134 [log10 equivalent] [13]. Given that TFU values were employed to calculate the predicted relative abundances, we can factor in a potential measurement uncertainty of ±0.10 in the log(TFU). Under this assumption, the standard range of variation for any measure is ~26% relative to the following:$${Relative\;Error=1-10}^{0.1}=1-1.2589\sim 26\%$$

Consequently, the absolute % error of the predicted values in a mix can be estimated as follows:$$Absolute\;\%\;Error=\frac{Relative\;Error}{N-1}=\frac{26}{5}\sim 5.2\%$$

Therefore, the observed average discrepancy of 6% between the predicted and actual observed values falls within a comparable error margin as expected by the FCC alone. This result means that the BifidoZoom test, when applied to a mixed sample, falls within the required measurement precision set forth in ISO 19344 and is therefore suitable for the quantification of *Bifidobacterium* in mixed starter cultures, probiotics, or fermentation. The calculated absolute error with an S_R_ of 0.134 and six elements in the mix would be ~7.2%.

### 3.2. Overview of Participants

For each of the 98 participants, stool samples, urine samples, and the answers to the KIDMED questionnaire were collected. The general characteristics of each of the groups can be seen in Table 2.

As can be seen from Table 2, there is no significant difference in average age between the three groups since they all fall within one standard deviation of one another. The ratio between males and females was found to be independent of the group by a χ^2^ test. The BMI is not significantly different between the T1D and control groups since they fall within one standard deviation of one another. As expected, the obese cohort has significantly larger BMI values.

### 3.3. Univariate Analysis

From a univariate analysis perspective, the relative abundance of different *Bifidobacterium* species was assessed in relation to obesity and T1D. It was noted that *B. bifidum* appeared less frequently in obese individuals, whereas *B. longum* subspecies *infantis* was significantly more prevalent. The fact that these significant differences are consistent when comparing obese individuals to both HCs and those with T1D adds weight to the findings (see Figure 2).

Furthermore, *B. pseudocatenulatum* shows a higher relative abundance in T1D participants in comparison to HCs. In terms of metabolites, distinct and significant variations were observed among the groups for several SCFAs and BCAAs. The most pronounced differences were between the obese and T1D subjects, underscoring the potential metabolic distinctions linked with these conditions.

Figure 3 shows the association between the KIDMED score and the different study groups. Surprisingly, the KIDMED score in obese children is higher than in the HCs, indicating that obese children had a healthier diet than the HCs. This may be because these children were already receiving dietary advice, or because of a reporting bias. No differences were observed between the HCs and the T1D group.

As can be seen in Figure 4 and in agreement with observations from the profiling of the gut microbiome, the analysis of SCFAs and BCAAs showed a significant shift in both study groups. For example, valeric acid is significantly higher in obese individuals compared to HCs, and again higher in the type 1 diabetes group compared to both other groups. Leucine and valine (BCAAs) are found to be significantly lower in the obese group than in both other groups (which are not significantly different), while leucine (the third BCAA) seems to be lower in obese individuals, but the significant decrease is only measured between the T1D and obese groups. In general, higher levels of most SCFAs were found in the urine of type 1 diabetics. These intergroup alterations in the levels of substrates (e.g., BCAAs) or metabolic products of the gut microbiome (e.g., SCFAs) indicate a potential change in the activity of the gut microbiome in obese and T1D children.

### 3.4. Multivariate Analysis

Figure 5 depicts the outcome of the PLS-DA applied to the microbiome data. It illustrates distinct clustering among the study groups, with notable demarcation between obese and control participants. The primary driver of this differentiation appears to be latent variable 1 (LV1), which shows a positive correlation of *B. breve* and *B. longum* subspecies *infantis* with obesity. In contrast, *B. longum* subspecies *longum* and *B. bifidum* are negatively correlated, suggesting an inverse relationship with obesity.

These PLS-DA results are in harmony with the univariate analysis findings, reinforcing the significance of *B. bifidum* and *B. infantis* in distinguishing obese individuals from HCs. An unexpected observation from the PLS-DA is the association of *B. breve* with obesity, a link that was not apparent in the univariate analysis. Additionally, the PLS-DA confirms the univariate analysis’s identification of *B. adolescentis* and its association with T1D, as well as a notable association with *B. pseudocatenulatum*, particularly evident in LV2.

To validate the PLS-DA model, one could investigate the correlation between LV1, as visualised in Figure 5, and the BMI of the participants. Such an analysis would help confirm whether the variations captured by LV1 accurately reflect the differences in BMI, thereby providing a robust methodological link between microbiome composition and obesity phenotypes.

Figure 6 shows a significant correlation between the BMI of participants and the LV that was detected using the PLS-DA. This shows that the LV computed based on the microbiome variables is a potential predictor of BMI.

Figure 7 illustrates the scores and loadings from the PLS-DA based on the metabolite concentrations found in urine samples from the HCs, obese subjects, and individuals with T1D. The PLS-DA model effectively distinguishes T1D subjects from HCs, as evidenced by the clear separation in the score plot.

The loadings for both LVs suggest that most of the variance in metabolites contributing to this separation pertains to differences between HCs and T1D subjects. This variance is particularly noticeable with changes in SCFA and BCAA levels between these two groups.

When we compare the findings of Figure 5, which reflects microbiome data, with those of Figure 7, there is a discernible pattern. The microbiome variables captured in Figure 5 predominantly account for the differentiation between obese individuals and controls. Conversely, the urine metabolites featured in Figure 7 provide a clearer understanding of the distinctions between HCs and T1D subjects. This complementary analysis from both figures highlights that while the microbiome variables are indicative of obesity, the urine metabolites are more reflective of the metabolic alterations associated with type 1 diabetes.

Figure 8 presents a correlation matrix for the measured variables, illustrating that the majority of metabolite concentrations are positively correlated with one another. The correlation shows in fact Spearman’s ρ, which is non-parametric. This finding aligns with the observations from Figure 7, where most metabolites showed positive loadings on LV1, indicating an association with T1D.

In the correlation matrix, significant associations emerge between certain microbiome constituents and metabolite concentrations. Notably, the levels of the amino acids valine and isoleucine, as well as the short-chain fatty acids valeric acid and isovaleric acid, exhibit the most robust correlations with the abundance of *Bifidobacterium* species. These strong associations suggest that the metabolic profiles, particularly the levels of these specific metabolites, are reflective of the microbial composition, further emphasising the interconnectedness of the microbiome and the host metabolism in the context of T1D.

## 4. Discussion

Our data show that individuals with obesity have a decrease in the relative abundance of *Bifidobacterium longum* subsp. *longum* and *Bifidobacterium bifidum* compared to healthy controls. These *Bifidobacterium* (sub)species have been previously associated with beneficial effects on gut health, metabolic processes, and weight management [18,19]. *B. longum* subs. *longum* has been shown to positively influence glucose and lipid metabolism, while *B. bifidum* has demonstrated potential in reducing body weight and improving insulin resistance. Our findings align with existing research and further emphasise the importance of these bacteria in maintaining a healthy weight.

Additionally, we observed reduced levels of isoleucine and valine in obese children in comparison to HCs. Isoleucine and valine are essential BCAAs that play crucial roles in energy metabolism and muscle protein synthesis [20]. Recent studies have highlighted the interplay between these BCAAs and the gut microbiome, particularly *Bifidobacterium* species. Isoleucine and valine have been found to promote the growth of bifidobacteria in the gut, creating a symbiotic relationship that supports metabolic health [21,22]. Conversely, a deficiency in these amino acids may lead to an imbalance in the gut microbiome, contributing to obesity and related metabolic disorders [23].

The metabolism of isoleucine and valine by *Bifidobacterium* species is a complex process that involves the breakdown of these amino acids into metabolites that can modulate gut health and systemic metabolism [24]. These species’ ability to utilise isoleucine and valine may enhance their colonisation in the gut, further supporting the host’s metabolic functions [25].

In type 1 diabetic children, the only significant difference in their *Bifidobacterium* profile compared to healthy controls is an increase in *B. adolescentis*. At the metabolic level, we observed some alterations in urine metabolites and, specifically, an increase in two SCFAs (propionic acid and butyric acid) as well as valeric and heptanoic acid.

It is interesting that the relative abundance of *B. longum* subs. *infantis* is increased in in the obese group. One could also interpret the results from Figure 2 and Figure 5 as meaning that *B. breve* is also increased in the obese group compared to the healthy controls. *B. infantis* is often cited as promoting weight gain early in life, a desirable effect [26]. It is however difficult to relate the impact of *B. infantis* in newborns to its possible role in obesity in teenagers, who rely on different metabolic pathways.

The partial least squares discriminant analysis visualised in Figure 5 and Figure 7 suggests that both metabolites and microbiome profiles are effective in differentiating between the three study groups. The delineation between obese subjects and HCs is more pronounced in Figure 5, which could be indicative of a stronger metabolic distinction. This highlights the potential of using *Bifidobacterium* as a biomarker for obesity and a target for dietary modification strategies.

Bifidobacteria composition reported by BifidoZoom in diabetic and obese children could be potentially exploited to elaborate personalised nutritional advice for those children. For example, in a nutritional intervention trial in T1D and obese children, a microbiome-targeted personalised diet was designed, with the aim of restoring *Bifidobacterium* species when they were selectively reduced in diabetic or obese children. These results are currently being analysed for further publication.

Sequencing-based microbiome assessments offer a broad overview of microbial populations within a sample, but they lack the specificity and quantification accuracy that targeted approaches provide. The BifidoZoom test, utilising a targeted analysis framework, is engineered to detect predetermined microbial species with high precision. This targeted detection is particularly advantageous in clinical contexts where the presence and abundance of specific microbial species could be used as a biomarker of specific health conditions. Our experiment, unlike classic metagenomic analyses, which catalogue the entire spectrum of microbiota present, focuses on the specific HRB for which it is designed. The validation studies corroborate that BifidoZoom’s precision in mixed samples is on par with the accuracy that flow cytometry achieves in pure samples, making it a formidable tool for clinicians and researchers seeking targeted microbiome insights.

The ATGC workflow, with PCR and capillary electrophoresis (CE), allows for significantly more control over the analytical process, such as internal standards in the CE. The validation data and the observations made show that ATGC is a valuable tool for routine clinical analysis of microbiome profiles for a well-defined taxonomic group. These well-defined taxonomic groups exist in specific microbiome niches, for example, the *Lactobacillus* genus in the vaginal microbiome [27], or the *Bifidobacterium* genus in the gut microbiome of newborns [27,28]. The observed link between bifidobacteria and obesity suggests that although there are several other genera present in the gut microbiome, the *Bifidobacterium* genus plays a key role. Routine clinical tests for precision supplementation could be based on specific microbial targets, rather than generic sequencing approaches. This conclusion is further supported by the results from other studies that found that *Bifidobacterium* is the genus that seems to play the most relevant role in obesity [29,30], with *B. longum* and *B. bifidum* being associated with protection from obesity [31].

Supplementation with *Bifidobacterium longum* subs. *longum* and *bifidum*, coupled with isoleucine and valine, could provide a targeted approach to mitigate the onset of obesity. Such interventions could harness the synergistic effects of these components to restore gut microbiome balance, enhance metabolic functions, and promote healthy weight management [32]. Furthermore, bifidobacteria’s ability to ferment dietary fibres into SCFAs contributes to a lower intestinal pH, fostering an environment conducive to beneficial bacteria [33,34]. This pH modulation may be particularly relevant in the context of obesity, where alterations in the gut microbiome and pH levels have been observed [35]. This hypothesis could be the foundation of further study.

The major limitation of our study is the limited number of cases; therefore, the results need further validation in a new study with a significantly higher number of recruited subjects. Furthermore, the number of participants in the obese group were smaller compared to the other groups due to difficulties in recruiting participants during the COVID-19 period.

In addition, although nutritional data were collected through KIDMED questionnaires, the results from their analysis were inconsistent and have therefore not been reported. The KIDMED scores were positively associated with obesity, meaning that according to its results, obese children had a healthier diet than HCs or those with T1D. This result could itself be significant, or the result of a reporting bias, and further investigation must be conducted to draw significant conclusions.

All results in this study are based on a young population. However, the same methodology could be expanded to adults. Studies suggest that the gut microbiome can be significantly different between children and adults [36,37]. Therefore, further studies would need to be carried out to understand if the same markers can be observed in adults as in children, or to identify the ones more relevant to specific age groups.

## 5. Conclusions

Microcapillary electrophoresis was used for the detailed profiling of *Bifidobacterium* species and subspecies focusing on cohorts with obesity and type 1 diabetes. When paired with targeted metabolomics, this approach has revealed associations between the abundance of *B. longum* subsp. *longum* and *B. bifidum* and the concentrations of isoleucine and valine in both healthy controls and children affected by obesity or T1D. Our study contributes to a nuanced understanding of the dynamic relationship between the gut microbiome and metabolic health. These results provide insights for the development of personalised nutrition strategies that utilise specific bacterial strains and amino acids to address childhood obesity. The potential of such a strategy to improve obesity prevention and management is substantial, offering a refined and evidence-based method to confront a global health issue. Future nutritional guidance is likely to be enhanced by advanced diagnostics that offer comprehensive, rapid, and accurate analysis of the gut microbiome, tailored to individual needs. Such tools will be crucial in translating our expanding knowledge of the gut microbiota and its metabolic interactions into practical clinical applications, thereby revolutionising the approach to maintaining metabolic health and preventing obesity.

## Figures and Tables

**Figure 1 microorganisms-12-00931-f001:**
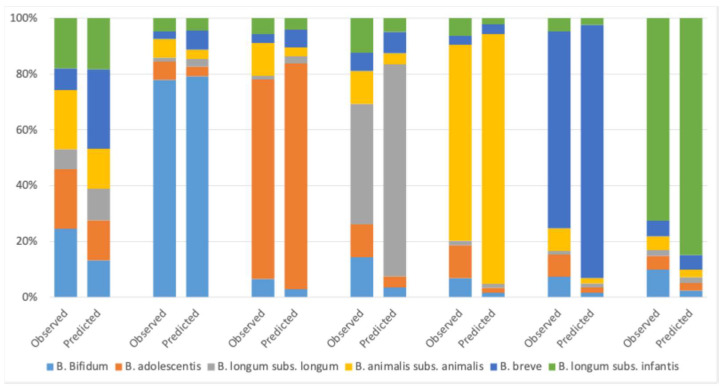
The results from the validation experiment for BifidoZoom. For each of the seven mixes, there is an observed value (BifidoZoom) and a predicted value (i.e., the result from calculating the relative abundance using weight and CFU/g as measured with flow cytometry).

**Figure 2 microorganisms-12-00931-f002:**
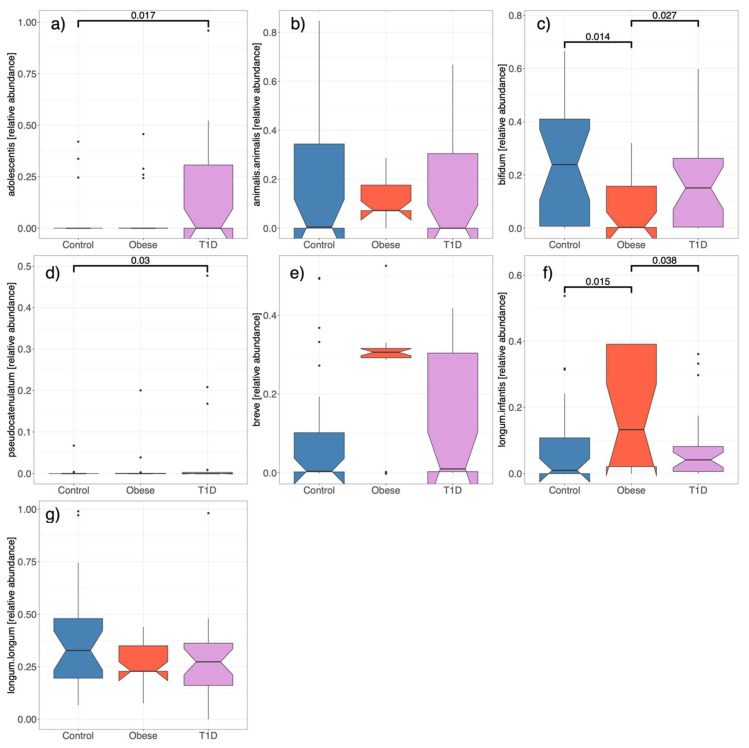
The relative abundance of specific *Bifidobacterium* species. The results are displayed as boxplots with brackets indicating significant differences between groups in paired comparisons. No bracket means no significant difference. *p*-values were computed using the Wilcoxon test. (**a**) *B. adolescentis*, (**b**) *B. animalis* subs. *animalis*, (**c**) *B. bifidum*, (**d**) *B. pseudocatenulatum*, (**e**) *B. breve*, (**f**) *B. longum* subs. *Infantis*, and (**g**) *B. longum* subs. *longum*.

**Figure 3 microorganisms-12-00931-f003:**
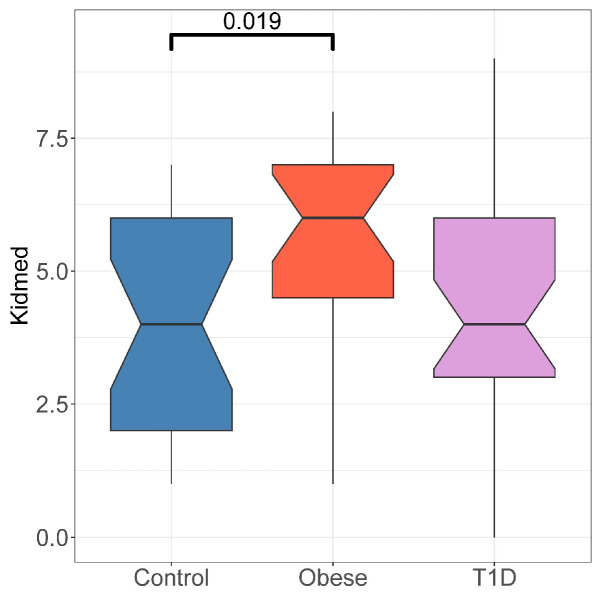
KIDMED scores displayed as a boxplot. Significant *p*-values are indicated. *p*-values were computed using the Wilcoxon test.

**Figure 4 microorganisms-12-00931-f004:**
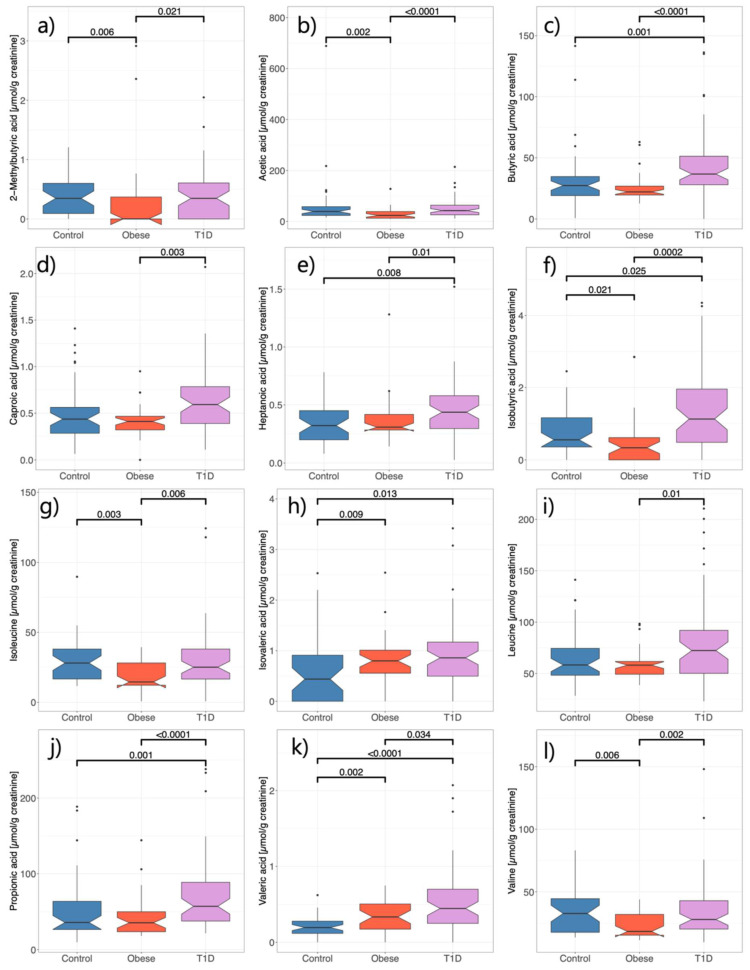
Normalised urinary concentrations of (**a**) 2-methylbutyric acid, (**b**) acetic acid, (**c**) butyric acid, (**d**) caproic acid, (**e**) heptanoic acid, (**f**) isobutyric acid, (**g**) isoleucine, (**h**) isovaleric acid, (**i**) leucine, (**j**) propionic acid, (**k**) valeric acid, and (**l**) valine, determined in the study cohort. Note: Significant *p*-values are indicated. *p*-values were computed using the Wilcoxon test.

**Figure 5 microorganisms-12-00931-f005:**
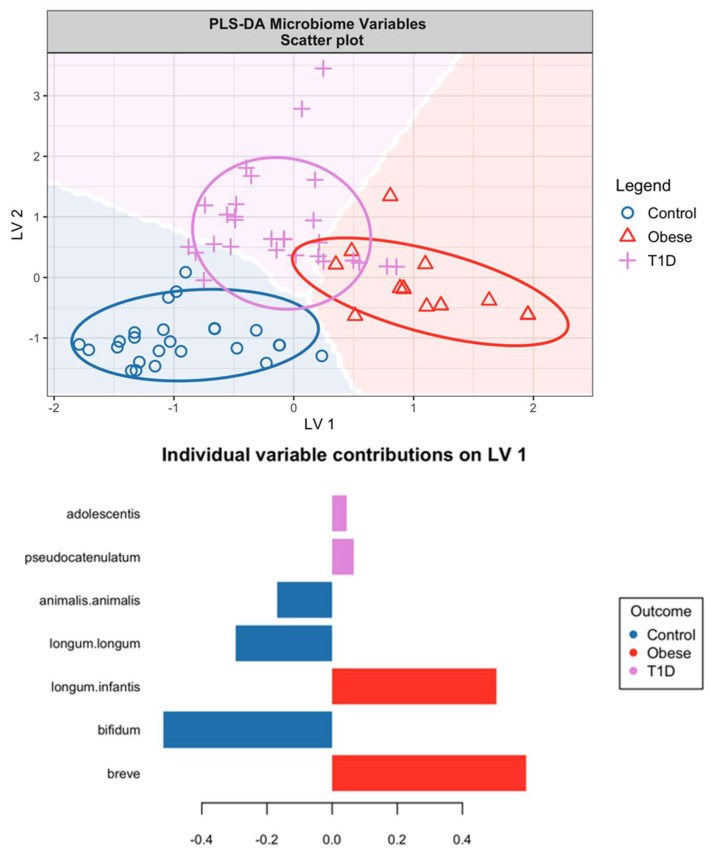
Score (**top**) and loading (**bottom 2**) plots from the PLS-DA based on gut microbiome profiles. The models used two latent variables (LVs) and all variables have been scaled to zero mean and unit variance.

**Figure 6 microorganisms-12-00931-f006:**
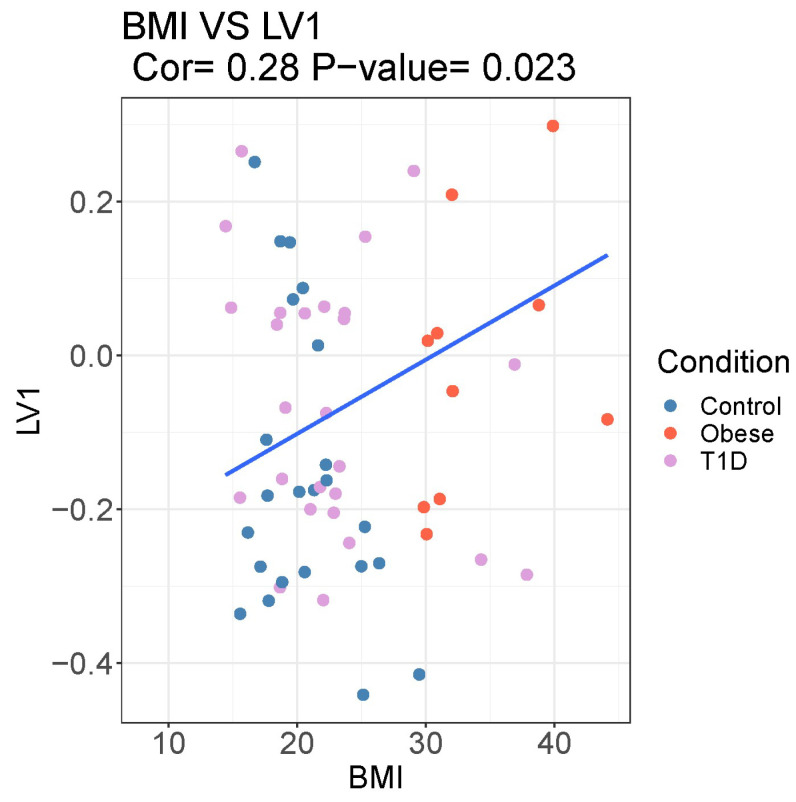
Scatterplot and correlation between BMI and the first latent variable (LV1) of the PLS calculation. The correlation is Spearman’s ρ, and the *p*-value is calculated using Spearman’s method.

**Figure 7 microorganisms-12-00931-f007:**
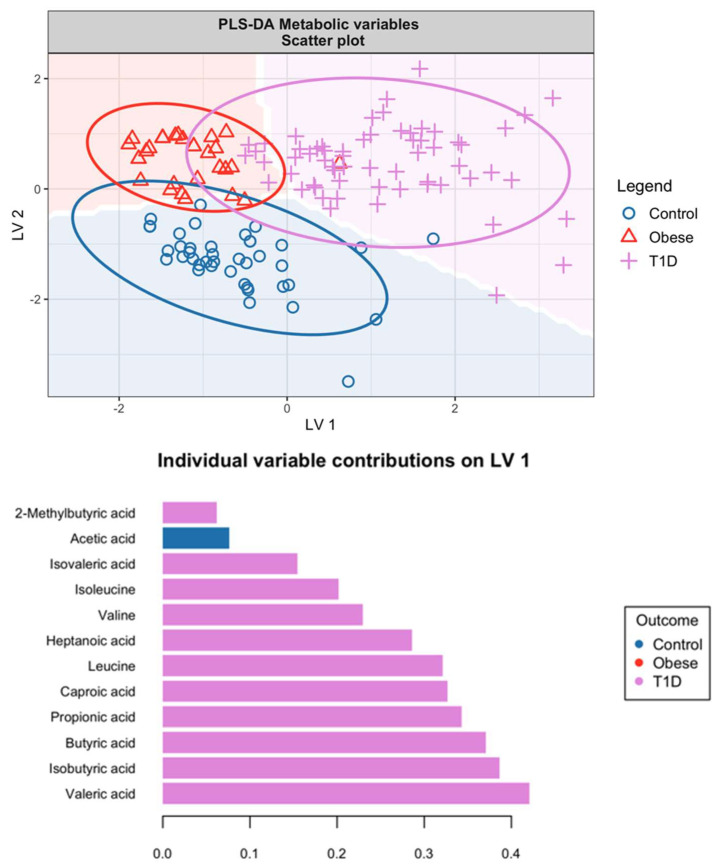
Score (**top**) and loading (**bottom 2**) plots from the PLS-DA based on urinary metabolite concentrations. The models used two latent variables (LVs) and all variables have been scaled to zero mean and unit variance.

**Figure 8 microorganisms-12-00931-f008:**
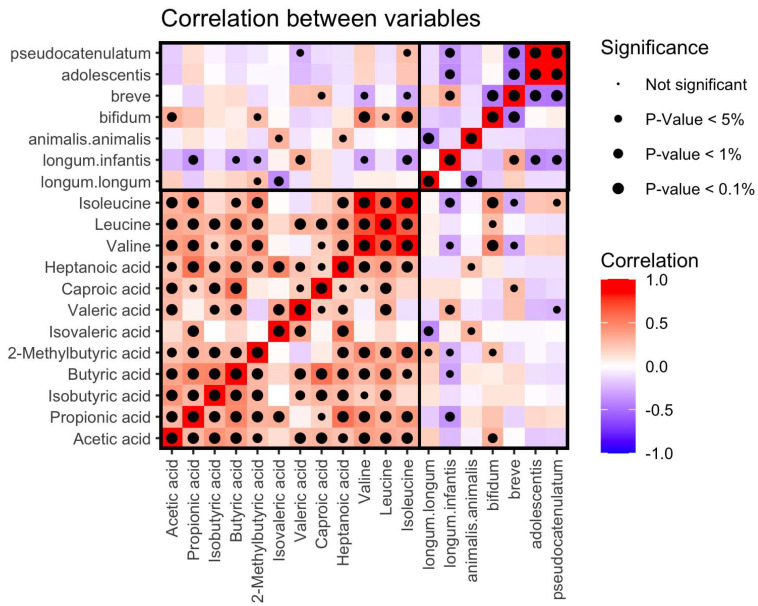
A tile plot showing the correlation between variables. The correlations were computed using Spearman’s method. The dots in the middle of each tile represent the level of significance measured using a two-sided Pearson test.

**Table 1 microorganisms-12-00931-t001:** Species of *Bifidobacterium* to which each of the strains used for the validation experiment belonged. All strains were manufactured by Probiotical S.p.A (Novara, Italy).

Species	*Adolescentis*	*Animalis* Subs. *Lactis*	*Bifidum*	*Breve*	*Longum* subs. *Infantis*	*Longum* Subs. *Longum*
Strain	BA02	BS01	BB10	BR03	BI02	BL03

**Table 2 microorganisms-12-00931-t002:** The general characteristics of the three groups. Age and BMI are shown with ±standard deviation. As can be seen, the ages are not significantly different. The BMI distribution is also similar between the controls and the T1D group. The ratio of males to females is also well conserved across all groups.

Group	Male	Female	M/F %	Age	BMI	HbA1C (%)
**Controls**	22	17	56.4%	12.67 ± 2.34	20.51 ± 3.59	
**Type 1 Diabetes**	25	15	62.5%	12.65 ± 2.59	22.15 ± 5.23	7.09 ± 2.01
**Obese**	8	10	44.4%	13.33 ± 2.4	32.46 ± 4.77	

## Data Availability

Data are contained within the article. A GitHub repository has been made public: https://github.com/pauloref/FrontierMicrobiologyBifidobacteriaFiguresAndData, accessed on 17 April 2024. It contains the complete dataset as well as an R notebook that generates all figures (except Figure 1).

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
