# Peer review of "Innovative Biomarkers for Obesity and Type 1 Diabetes Based on Bifidobacterium and Metabolomic Profiling"

_microorganisms, 2024, doi:10.3390/microorganisms12050931_

Round 1
Reviewer 1 Report
Comments and Suggestions for Authors
Nobili et al. Innovative Biomarkers for Obesity and Type 1 Diabetes Based on Bifidobacterium and Metabolomic Profiling. The study recruited groups of children with different health conditions, including obesity, type 1 diabetes, and healthy controls, which facilitated comparative analysis of gut microbiota and metabolite profiles in different disease states. High-resolution quantitative analysis of bifidobacteria in the intestine was performed using the innovative “Bifido Zoom” technique, combined with metabolomic analysis, which laid the foundation for the discovery of biomarkers of obesity and type 1 diabetes. The result is technically sounded. However, there are still some comments and suggestions that are given to improve the manuscript:
1. There is a lack of analysis of study limitations in the discussion. For example, the sample size is relatively small, especially only 18 in the obese group, which may not be fully representative of this population. The sample size could be expanded in the future to enhance the reliability of the conclusions.
2. When discussing the potential of individualized nutritional interventions, it is possible to explore more deeply how to guide clinical practice based on the Bifidobacterium microbiota and metabolite profiles found in this study. For example, individualized nutritional intervention strategies for different disease states can be discussed.
4. The tables are not aesthetically pleasing. In particular, the data advantages of the table are messy. It is recommended to adjust the appropriate length to present a better display. It is appropriate to adjust the tables to a three line format.
5. The figures are not aesthetically pleasing. In particular, the font sizes in the picture are different. It is also recommended to adjust the appropriate length to present a better display.
6. Further statistical analysis is needed to determine if there is a significant difference in the patient's general characteristics.
7. In Figure 2, the KIDMED score in obese children is higher than in the HC (p=0.019). Is there a significant difference in KIDMED scores between TiD children and HC children?
8. In Figure 3 and 4, it is appropriate to add sub labels such as “a,b,c,d”to each figure and explain them in the legend.
Comments on the Quality of English LanguageModerate editing of English language required
Author Response
Dear Reviewer,
Thank you for your positive response, and constructive comments to improve our manuscript.
Please see the attachments. You will find a point-by point response to your comments, as well as a reviser manuscript.
Thank you very much
Best regards
Paulo Refinetti

Reviewer 2 Report
Comments and Suggestions for Authors
Although your study was conducted in children the potential benefit in adults is promising. What results have you generated in adults? Ameliorating obesity is a worldwide challenge.Metabolic syndrome and steatohepatitis is becoming a common cause of cirrhosis and a spectrum of complications including hepatocellular carcinoma.
The methodology here is tapered for infants and if this is applied to adults where it could be helpful in managing obesity those studies should be performed.
The controls should be a spectrum of other biomarkers found in stools in non-obese individuals.
The authors point out that future nutritional guidance should be comprehensive, rapid, and involve individual needs which are not addressed in this pediatric study.
This is an important study but to apply this to adults with obesity and in particular steatohepatitis and cirrhosis adult populations need to be studied.
The references are important but ideally should include a reference on stem cells that unquestionably will be employed in the future therapeutic studies involving obesity and steatohepatitis.
The tables and figures are well done and very helpful. No need for changes.
Author Response

(The authors gave the same response as above.)
